# Promiscuous binding by Hsp70 results in conformational heterogeneity and fuzzy chaperone-substrate ensembles

Rina Rosenzweig[1,2,3,4*†], Ashok Sekhar[1,2,3*†], Jayashree Nagesh[5], Lewis E Kay[1,2,3,6*]

[1]Department of Molecular Genetics, The University of Toronto, Toronto, Canada; [2]Department of Biochemistry, The University of Toronto, Toronto, Canada; [3]Department of Chemistry, University of Toronto, Toronto, Canada; [4]Department of Structural Biology, Weizmann Institute of Science, Rehovot, Israel; [5]Chemical Physics Theory Group, Department of Chemistry, University of Toronto, Toronto, Canada; [6]Hospital for Sick Children, Program in Molecular Structure and Function, Toronto, Canada

*For correspondence: rina.
rosenzweig@weizmann.ac.il (RR);
ashoksekhar@pound.med.
utoronto.ca (AS); kay@pound.
med.utoronto.ca (LEK)

†These authors contributed
equally to this work

Competing interest: See
page 18

Reviewing editor: Volker
Dötsch, JW Goethe-University,
Germany

**Abstract** The Hsp70 chaperone system is integrated into a myriad of biochemical processes that are critical for cellular proteostasis. Although detailed pictures of Hsp70 bound with peptides have emerged, correspondingly detailed structural information on complexes with folding-competent substrates remains lacking. Here we report a methyl-TROSY based solution NMR study showing that the *Escherichia coli* version of Hsp70, DnaK, binds to as many as four distinct sites on a small 53-residue client protein, hTRF1. A fraction of hTRF1 chains are also bound to two DnaK molecules simultaneously, resulting in a mixture of DnaK-substrate sub-ensembles that are structurally heterogeneous. The interactions of Hsp70 with a client protein at different sites results in a fuzzy chaperone-substrate ensemble and suggests a mechanism for Hsp70 function whereby the structural heterogeneity of released substrate molecules enables them to circumvent kinetic traps in their conformational free energy landscape and fold efficiently to the native state.

## Introduction

The Hsp70 chaperone system plays a vital role in quality control by overseeing the integrity of the cellular proteome in organisms ranging from bacteria to humans (*Balchin et al., 2016*). Hsp70 performs a staggering array of functions and its ATP-dependent interaction with client proteins imparts directionality to a number of cellular processes such as protein folding (*Hartl et al., 2011*), translocation (*Kang et al., 1990*; *Pilon and Schekman, 1999*; *Mayer, 2013*), disaggregation (*Nillegoda et al., 2015*; *Mogk et al., 2015*) and oligomer disassembly (*Böcking et al., 2011*; *De Los Rios and Goloubinoff, 2016*). Hsp70 is able to interact with clients having a broad range of conformations (*Schlecht et al., 2011*; *Mashaghi et al., 2016*), including unfolded polypeptides emerging from the ribosome (*Teter et al., 1999*; *Deuerling et al., 1999*; *Willmund et al., 2013*), native proteins such as the heat shock transcription factor (*Rodriguez et al., 2008*), macromolecular assemblies including clathrin-coated vesicles (*Sousa et al., 2016*), as well as non-native misfolded and aggregated polypeptide chains (*Szabo et al., 1994*; *Mayer and Bukau, 2005*; *Goloubinoff and De Los Rios, 2007*).

The multitude of client conformations that Hsp70 recognizes and the diversity of functions it carries out are in sharp contrast to the simplicity of its client binding-site architecture (*Clerico et al., 2015*). Hsp70 interacts with client substrates through its C-terminal domain, which is composed of a

β-sandwich subdomain that includes a substrate binding cleft and an α-helical lid (*Figure 1A*). Both crystal- (*Zhu et al., 1996*; *Zahn et al., 2013*) and NMR-derived structures (*Stevens et al., 2003*) show that short peptides bind Hsp70 in an extended conformation with only five residues contacting

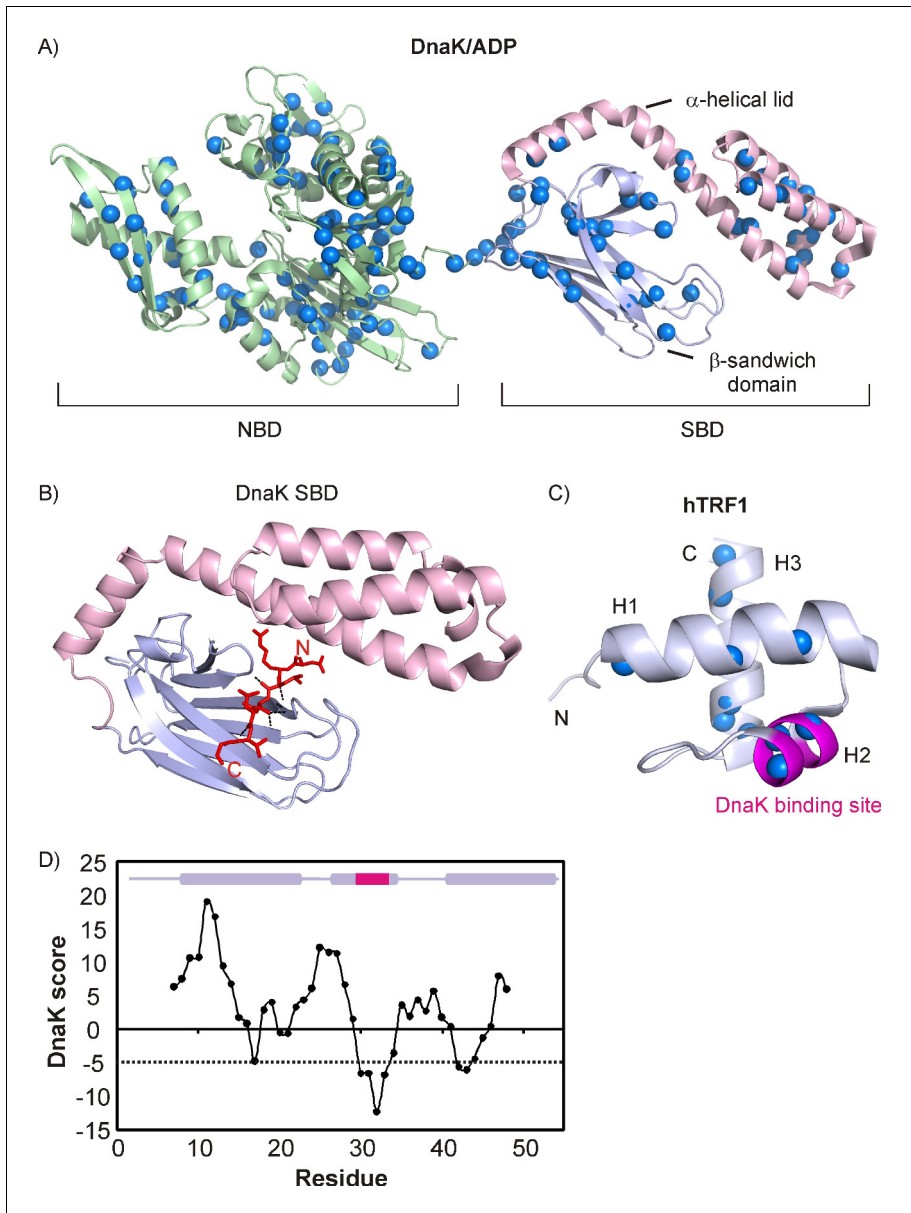

**Figure 1.** Architecture of DnaK and hTRF1 molecules. (**A**) Cartoon representation of DnaK/ADP (PDB ID: 2KHO [*Bertelsen et al., 2009*]) highlighting the nucleotide binding (green, NBD) and substrate binding (SBD) domains, the latter of which is further subdivided into the β-sandwich domain (steel blue) and the α-helical lid (light pink). The backbone nitrogen atoms of Ile, Leu, Val and Met residues are indicated as blue spheres. (**B**) Crystal structure of the NRLLLTG peptide in complex with the DnaK SBD (PDB ID: 1DKZ [*Zhu et al., 1996*]) depicting the extended conformation of the peptide. Hydrogen bonds between the peptide backbone and the β-sandwich domain are shown as black dashed lines. (**C**) Cartoon representation of the three-helix bundle hTRF1 (PDB ID: 1BA5 [*Nishikawa et al., 1998*]) showing the strongest DnaK binding site in magenta. The backbone nitrogen atoms of Ile, Leu, Val and Met residues are indicated as blue spheres. (**D**) The DnaK score for hTRF1 showing the presence of one strong (around Leu 30) and two weak DnaK binding sites (around Val 18 and Val 41). The secondary structure of hTRF1 is indicated on the top of the plot with the strongest DnaK binding site in magenta. A DnaK score of −5 denotes the cutoff for a binding site (*Rüdiger et al., 1997*) and is drawn as a dashed line.

Hsp70. The peptide is stabilized in the binding pocket by hydrogen bonds between its backbone and binding cavity forming loops in Hsp70, as well as by hydrophobic interactions between the peptide core and conserved Ile, Leu and Met residues of the binding cleft (*Figure 1B*). Crystal structures of a number of peptides bound to Hsp70 have been solved with Ile, Leu, Pro and the unnatural amino acid cyclohexylalanine occupying the central position of the binding pocket, suggesting that the pocket may possess considerable intrinsic flexibility (*Zahn et al., 2013*).

The ability of Hsp70 to restructure the conformation of its clients underlies many of its functions; however, the mechanism by which Hsp70 interacts with full-length proteins to modulate their conformation remains poorly understood. A number of models have been proposed in this regard, ranging from the active induced fit-like unfoldase model to the passive conformational selection-based holdase model (*Mayer, 2013*; *Goloubinoff and De Los Rios, 2007*; *Slepenkov and Witt, 2002*). During the course of its chaperone cycle, Hsp70 samples ATP-bound, ADP-bound and nucleotide-free states and also undergoes massive structural reorganization (*Kityk et al., 2012*; *Qi et al., 2013*; *Bertelsen et al., 2009*). These changes in Hsp70 structure have been hypothesized to perform conformational work on the bound substrate (*Mayer, 2013*), thereby changing its conformation as in the case of luciferase refolding, where Hsp70 is pictured as an unfoldase that converts a kinetically trapped misfolded state of luciferase into a folding-competent unfolded conformation (*Sharma et al., 2010*). In contrast, Hsp70 is believed to uncoat clathrin pits through a holdase mechanism by capturing conformers that are transiently formed via fluctuations in the clathrin structure (*Böcking et al., 2011*).

We have previously characterized how binding of Hsp70 affects local structure and long-range interactions in regions removed from the binding site of a helical protein model substrate, the human telomeric repeat-binding factor (hTRF1, *Figure 1C*) (*Sekhar et al., 2015*, *2016*). Notably, the conformation of the bound substrate is independent of the nucleotide state of Hsp70, suggesting that an ATP-coupled unfoldase mechanism may not be a general feature of Hsp70 activity (*Sekhar et al., 2015*). Hsp70 binding does not disturb the secondary structural propensities of hTRF1 removed from the binding site, although long-range interactions are significantly affected (*Sekhar et al., 2016*). Here, we have used methyl-TROSY NMR (*Tugarinov et al., 2003*) in conjunction with selective isotope labeling (*Tugarinov and Kay, 2004*; *Kerfah et al., 2015*) to obtain atomic resolution information on an hTRF1 complex with the major *E.coli* Hsp70 homolog, DnaK, including at the DnaK binding site. We find that the flexibility inherent in the DnaK binding pocket enables it to interact promiscuously with substrate. DnaK binds at as many as four distinct sites on this small 53-residue client protein, with a fraction of hTRF1 bound to two molecules of chaperone, resulting in considerable heterogeneity in the bound ensemble. Our results, therefore suggest a model for Hsp70-mediated conformational biasing whereby binding of Hsp70 at different positions of a substrate creates sub-ensembles with distinct conformational preferences and hence different starting structures for folding upon release from Hsp70, thereby broadening the conformational space sampled by the substrate as it folds. More generally, our study establishes that the DnaK-hTRF1 ensemble is a fuzzy complex, characterized by both static and dynamic disorder. It further demonstrates the utility of NMR spectroscopy as a tool to probe heterogeneous biomolecular ensembles at atomic resolution.

## Results

### A strategy for characterizing the interaction of DnaK with a model substrate using methyl-TROSY NMR

In order to obtain insight into the molecular details of Hsp70-substrate interactions we have used the model substrate hTRF1 (*Figure 1C*), a 53-residue three-helix bundle belonging to the homeodomain family of proteins (*Gianni et al., 2003*). hTRF1 is only marginally stable and its native conformation is in equilibrium with an unfolded state that is populated at 4.2% and only transiently formed with a lifetime of 2.9 ms (35°C) (*Sekhar et al., 2015*). This substrate has one predicted (*Rüdiger et al., 1997*) strong DnaK binding site centred at Leu 31 and two additional weak sites around residues 16 and 42 (*Figure 1D*). We have previously shown that hTRF1 interacts with DnaK at its canonical binding site and while DnaK-bound hTRF1 is globally unfolded, secondary structural propensities are not affected in substrate regions distal to the DnaK binding site, with as much as

40% residual secondary structure retained (*Sekhar et al., 2015*). In contrast, a series of spin relaxation experiments established that transient long-range native and non-native tertiary interactions that are present in the unfolded state of hTRF1 are disrupted in the bound form (*Sekhar et al., 2016*). Interestingly, despite the fact that DnaK undergoes large conformational changes between ATP and ADP bound states the tethered hTRF1 is maintained in a globally unfolded conformation throughout the entire chaperone nucleotide cycle (*Sekhar et al., 2015*).

Our initial studies using $^1$H-$^{15}$N NMR spectroscopy could not detect hTRF1 residues located at or near the DnaK binding site (residues 27–39 were invisible), because the relevant $^1$H-$^{15}$N correlations were not visible in TROSY-based spectra of U-$^2$H/$^{15}$N hTRF1 bound to U-$^2$H DnaK/ADP that reflects both the slow rotational tumbling of the ~80 kDa DnaK-hTRF1 complex and potentially also conformational heterogeneity at the binding site (see below). In order to probe chaperone-substrate interactions at the binding site we made use of selective isotope labeling strategies (*Kerfah et al., 2015*; *Tugarinov et al., 2006*) in concert with methyl Transverse Relaxation Optimized Spectroscopy (methyl-TROSY) (*Tugarinov et al., 2003*; *Ollerenshaw et al., 2003*). The methyl-TROSY technique takes advantage of favorable relaxation properties of the $^{13}$C and $^1$H methyl spins during magnetization transfer in a heteronuclear multiple quantum correlation (HMQC) experiment, where one of the two magnetization transfer pathways is effectively immune to intra-methyl dipolar relaxation (*Tugarinov et al., 2003*). External sources of transverse relaxation are minimized by preparing highly deuterated proteins, with $^1$H,$^{13}$C labeling restricted to the methyl groups of Ile, Leu, Val and Met and where the isopropyl methyls of Leu and Val are labeled as $^{13}$CH$_3$/$^{12}$CD$_3$. The utility of methyl $^{13}$CH$_3$-labeling along with methyl-TROSY methodology has been established in studies of high molecular weight proteins where insight into the structure and dynamics of a number of large molecules and macromolecular complexes with molecular weights up to approximately one MDa have been obtained (*Rosenzweig and Kay, 2014*; *Wiesner and Sprangers, 2015*; *Huang and Kalodimos, 2017*).

## Multiple DnaK-bound conformations of hTRF1

Human TRF1 has a number of methyl groups that can serve as probes of molecular structure and dynamics in the DnaK-bound complex. Of the single Ile, six Leu, two Val and two Met residues, the Ile and two of the Leu are located at the strongest DnaK binding site (*Figure 1C,D*). We have, thus, prepared a sample of U-$^2$H, Ileδ1-$^{13}$CH$_3$, Leu,Val-$^{13}$CH$_3$/$^{12}$CD$_3$, Met-$^{13}$CH$_3$-(referred to as ILVM-$^{13}$CH$_3$) labeled hTRF1 and exploited the methyl-TROSY effect to obtain spectra of high resolution and sensitivity. The Ile region of the $^1$H-$^{13}$C HMQC spectrum of ILVM-$^{13}$CH$_3$ hTRF1 shows a single peak arising from Ile 29, located in helix 2 (*Figure 2A*). However, upon addition of 2-fold excess U-$^2$H DnaK/ADP, at least five peaks are observed for Ile 29 at chemical shifts that are distinct from the resonance position of Ile 29 in the free state, labeled as A-E in *Figure 2B*, suggesting that there are multiple conformations of DnaK-bound hTRF1. In order to ensure that all the new Ile peaks arise from chaperone-bound hTRF1 molecules, we measured peak-specific diffusion coefficients using a 2D $^1$H-$^{13}$C HMQC-based pulsed field gradient (PFG) diffusion experiment. Of the diffusion coefficients that could be quantified reliably, those measured from the new peaks are very similar and are smaller than the free hTRF1 diffusion rate (*Figure 2—figure supplement 1*), establishing the existence of multiple DnaK-bound states of hTRF1 and thus conformational heterogeneity in the bound ensemble. The multiplicity of conformations that are reported by the Ile probes is further confirmed by the Leu/Val region of $^1$H-$^{13}$C HMQC spectra (as discussed in detail below). The majority of Leu and Val residues in hTRF1 are expected to be far removed from the strongest binding site and thus in the globally unfolded DnaK-bound state their associated methyl groups would be localized to the random coil region of HMQC spectra. However, there are a number of peaks outside this region (*Figure 2C*, labeled as 1–6) which, as described subsequently, arise from hTRF1 methyl groups of residues located directly at the DnaK binding site.

The conformational heterogeneity of the chaperone-substrate ensemble can be detected from spectra of DnaK as well. Upon addition of 2-fold excess U-$^2$H hTRF1 several new peaks appeared in the $^1$H-$^{13}$C HMQC spectrum of IM-$^{13}$CH$_3$ labeled DnaK/ADP (compare *Figure 2D and E*). These could be assigned via a methyl-NOESY data set recorded on an ILVM-$^{13}$CH$_3$ DnaK/ADP-U-$^2$H TRF1 sample, making use of available assignments for many of the methyl groups of DnaK/ADP (*Zhuravleva et al., 2012*), as arising from four separate conformers of Ile 401 and Ile 438. The resulting crosspeaks are labeled as 'a'-'d' in *Figure 2E* along with the Ile residue from which they derive.

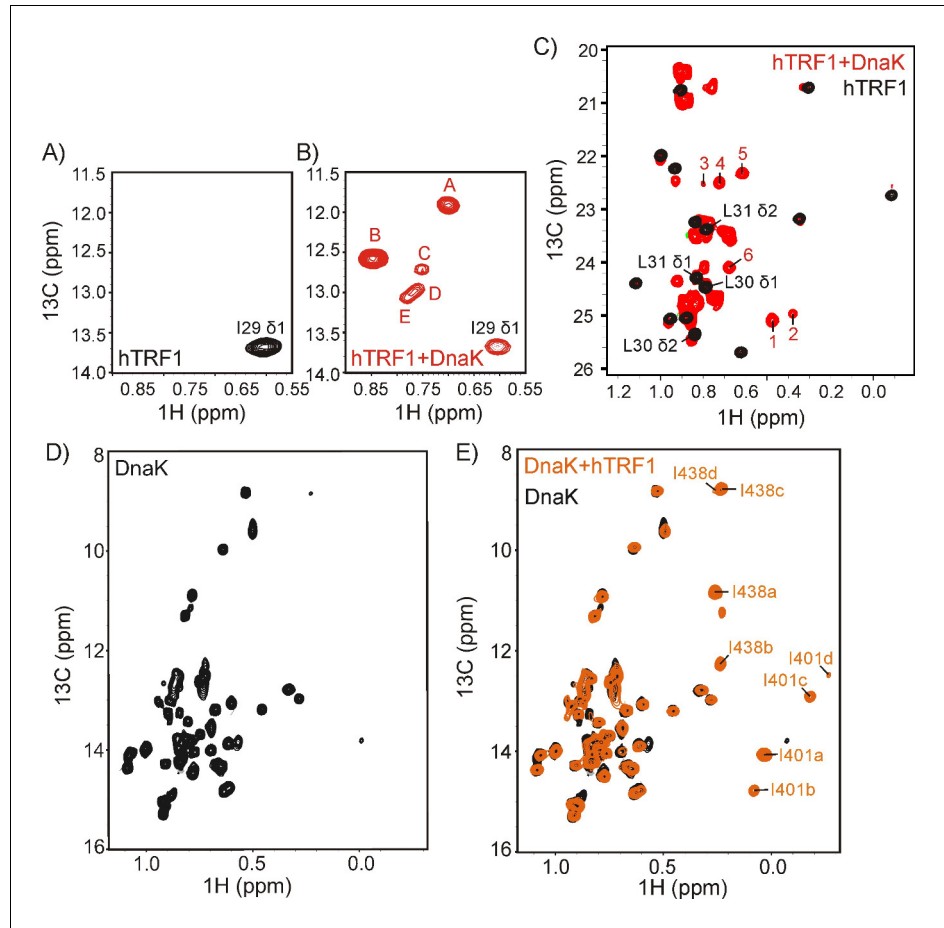

**Figure 2.** DnaK binds substrate via a multiplicity of interactions. The Ile region of $^1$H-$^{13}$C HMQC spectra of ILVM-$^{13}$CH$_3$ hTRF1 in the absence (**A**) and in the presence (**B**) of two-fold excess U-$^2$H DnaK/ADP. The single Ile 29 peak in the unbound state (**A**) disperses into multiple resonances in the hTRF1-DnaK bound state, indicating multiple DnaK-bound conformations of hTRF1. These are denoted by the letters A-E. (**C**) Superposition of the Leu/ Val regions of the same spectra shown in panels A and B with peaks arising from binding to DnaK that are outside the random coil region labeled from 1 to 6. (**D,E**) $^1$H-$^{13}$C HMQC spectra of IM-$^{13}$CH$_3$ DnaK/ADP in the absence (**D**) and in the presence of two-fold excess U-$^2$H hTRF1 (**E**) showing multiple peaks arising from Ile 401 and Ile 438 of hTRF1-bound DnaK/ADP and labeled as 'a'-'d' along with the Ile residue from which the peak originates.

The following figure supplements are available for figure 2:

**Figure supplement 1.** Verifying binding with a PFG diffusion experiment.

**Figure supplement 2.** Multiple conformers are observed upon DnaK binding to hTRF1 peptides.

In order to estimate the lifetimes of bound states we recorded a magnetization exchange experiment (*Farrow et al., 1994*) where the appearance of exchange-relayed crosspeaks between HMQC correlations provides evidence of exchange. No such peaks were observed in experiments acquired with a mixing time of 300 ms, establishing that the interconversion between multiple bound states is slow. Simulations that we have performed are consistent with a substrate off-rate, $k_{off}$, less than 0.3 s$^{-1}$. The absence of exchange between bound conformers is consistent with previously measured slow rates of substrate release from DnaK/ADP (~0.001 s$^{-1}$, 25°C) (*Pierpaoli et al., 1998*; *Mayer et al., 2000*), and in turn also establishes that if DnaK slides between different binding sites on a substrate polypeptide chain (see below) the rate must be on the order of 0.3 s$^{-1}$ or slower as well.

## Characterizing the multiple conformations of the DnaK-hTRF1 complex

Having established that the substrate-DnaK complex is conformationally heterogeneous, we proceeded to characterize the multiple bound states in more detail. Initially we wondered as to whether the entire hTRF1 polypeptide is essential to observe a multiplicity of bound states or if the heterogeneity can be reproduced by the region of the substrate alone that corresponds to the DnaK binding site. We considered the possibility that regions of hTRF1 removed from the binding site could interact with the DnaK chaperone in different ways, leading to the extra peaks in spectra of both the substrate and the chaperone. In order to address this possibility, we synthesized a small peptide of 16 residues, $hTRF1_{23-38}$, that includes the strongest predicted DnaK binding site but lacks the surrounding region and acquired $^1H$-$^{13}C$ HMQC spectra of IM-$^{13}CH_3$ DnaK/ADP in the free and peptide-bound states (*Figure 2—figure supplement 2A*, left). Notably, extra peaks for DnaK corresponding to Ile 401a,b and Ile 438a,b were observed so that the presence of the entire substrate polypeptide chain is not a pre-requisite for observing multiple bound conformations. In a similar manner a second peptide, $hTRF1_{34-49}$, was produced that contains a weaker DnaK binding site. An extra peak corresponding to Ile 401 c is observed in this case (*Figure 2—figure supplement 2A*, right) that is distinct from the peaks detected in the spectrum of IM-$^{13}CH_3$ DnaK/ADP bound to $hTRF1_{23-38}$ (*Figure 2—figure supplement 2A*, left). These experimental results are inconsistent with the notion that the extra conformers reflect multiple potential interactions between regions of the substrate distal from the binding site and DnaK.

Next, we tested whether the multiple conformations arise from the interactions between substrate and the α-helical chaperone lid that encloses the substrate-binding pocket. The lid has been hypothesized to adopt different positions relative to the binding cavity and each position could result in a distinct orientation for the hydrophobic sidechains of both the substrate and of DnaK (Ile 401, Ile 438) that line the binding site (*Schlecht et al., 2011*). In order to test this possibility, we carried out experiments with the ATP-bound form of DnaK as a way to modulate substrate-SBD-lid interactions and compared the results with those obtained from similar experiments recorded on DnaK/ADP. Substrate binding to DnaK/ATP results in the formation of an allosterically active DnaK ensemble that consists of domain-docked as well as -undocked conformations (*Zhuravleva et al., 2012*) in which the helical lid is more mobile than in the DnaK/ADP state, as inferred from faster substrate release rates in the ATP-bound conformation (*Mayer and Bukau, 2005*; *Gisler et al., 1998*). HMQC spectra of hTRF1 bound to DnaK/ATP were recorded using an ATP hydrolysis-deficient variant of the chaperone in which Thr at position 199 was mutated to Ala. The Ile region of the $^1H$-$^{13}C$ HMQC spectrum of hTRF1 bound to DnaK/ATP shows three separate resonances in addition to the peak from folded (unbound) hTRF1, *Figure 2—figure supplement 2B*, with the Leu/Val region showing additional resonances as well. This is similar to what was observed for the DnaK/ADP bound state where the lid is in the closed or down orientation, highlighting that multiple DnaK-bound states of hTRF1 are present even when the lid is not interacting (or interacting less) with substrate. These results thus argue against the lid being the source of the observed conformational heterogeneity.

High-resolution structures of DnaK-substrate complexes (*Zhu et al., 1996*; *Zahn et al., 2013*; *Stevens et al., 2003*) establish that the client protein is partly stabilized by hydrophobic interactions between its sidechains and those of the chaperone that line the binding cavity (*Figure 3A*). One substrate sidechain, designated to be at position 0 with respect to the DnaK binding cleft, extends deep into the hydrophobic pocket of DnaK and is in close proximity (~4 Å) to the δ1-methyl groups of Ile 401 and Ile 438 of the chaperone (*Figure 3A*). In order to ascertain whether different sidechains of the substrate occupy position 0, leading to the observed multiplicity of bound conformations, we used an isotope-labeling strategy which allowed us to simultaneously detect the key residues in both substrate and in DnaK. This was achieved by labeling the substrate at Ile γ2 and at the *proR*-positions of Leuδ and Valγ methyl groups (referred to as Iγ-*proR*-LV-$^{13}CH_3$) and labeling DnaK at Ileδ1 and Metε. *Figure 3B* (center) shows a $^1H$-$^{13}C$ HMQC spectrum recorded on a sample containing 1.8 mM DnaK/ADP and 0.7 mM hTRF1 incorporating the above labeling scheme. The crosspeaks from hTRF1 (black) are distinct in their $^{13}C$ chemical shifts from those belonging to DnaK (green).

We then exploited the proximity of Ile 401 and Ile 438 of DnaK to the corresponding substrate sidechain at position 0 that allowed us to link sets of peaks from hTRF1 (labeled 1–6) and DnaK (labeled 'a'-'d') deriving from the same conformation using a 3D $^{13}C$-edited NOESY experiment

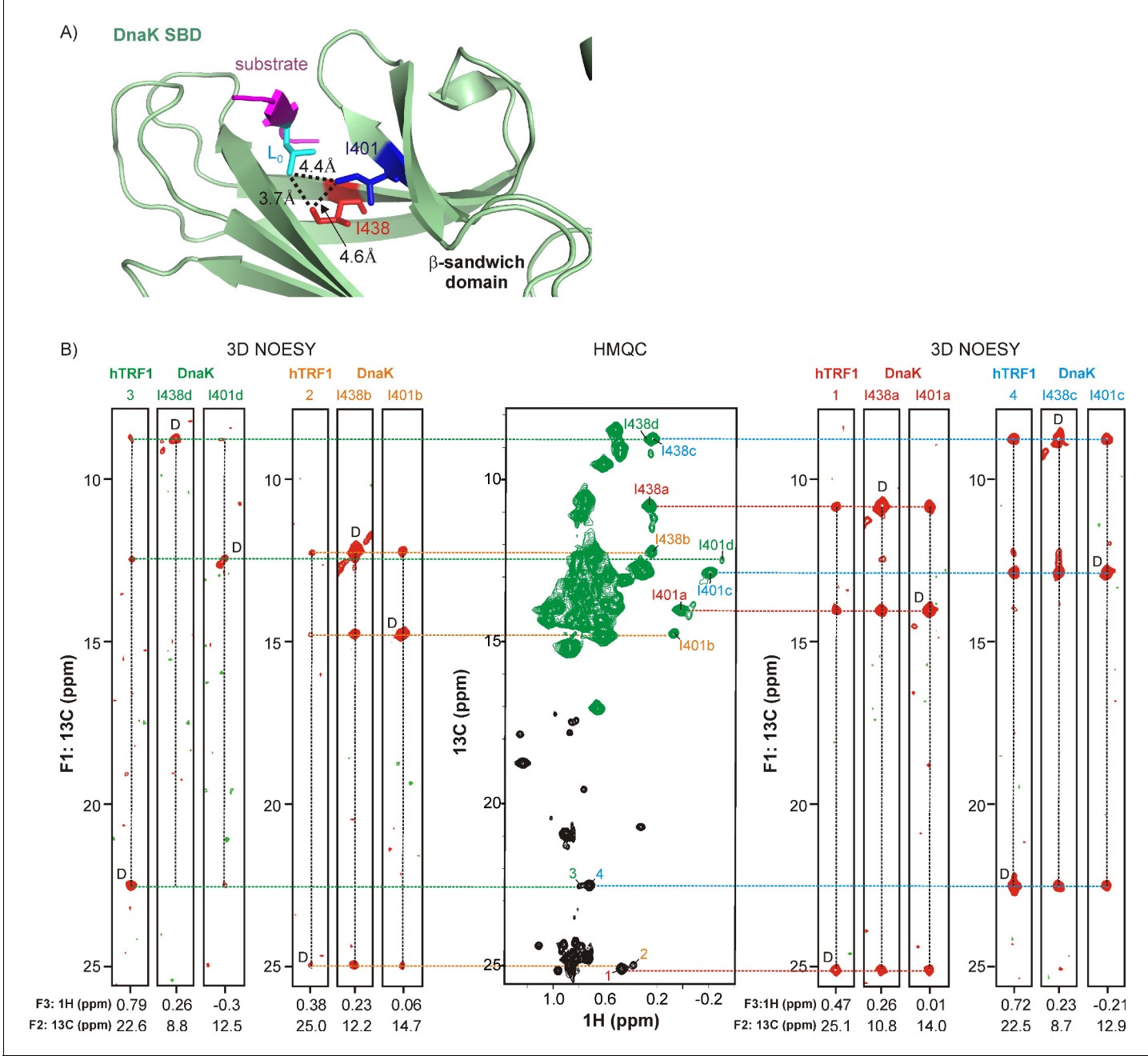

**Figure 3.** Linking multiple resonances and hence conformers of DnaK and hTRF1. (**A**) Zoomed region of the crystal structure of a DnaK-NRLLLTG peptide complex (PDB ID: 1DKZ [*Zhu et al., 1996*]) highlighting interactions between sidechains of the peptide (magenta) bound to the substrate binding domain (SBD) of DnaK (green). Key hydrophobic sidechains lining the central cavity of the DnaK binding pocket, Ile 401 and Ile 438, are shown as sticks and coloured blue and red respectively. The Leu residue in the substrate occupying this central cavity ($L_o$) and forming hydrophobic interactions involving Ile 401 and Ile 438 is shown in cyan. Distances between the key residues in the binding cavity are denoted on the plot. (**B**) (middle) $^1H$-$^{13}C$ HMQC spectrum of 0.7 mM I$\gamma$-*proR*-LV-$^{13}CH_3$ labeled hTRF1 bound to 1.8 mM IM-$^{13}CH_3$ labeled DnaK/ADP. Resonances in the HMQC spectrum arising from hTRF1 are coloured black, while peaks from DnaK are in green. Peaks are identified as in *Figure 2*. The HMQC dataset is flanked by four sets of $F_1$-$F_3$ strips from a 3D methyl NOESY spectrum with correlations of the form ($^{13}C_j$-NOE-$^{13}C_k$-$^1H_k$). The first set of strips link hTRF1 conformation 3 with DnaK conformation d, the second set link hTRF1 and DnaK conformations 2 and b and so forth. The letter 'D' is used to identify the diagonal peak in each plot. The $F_2$ $^{13}C$ chemical shift is indicated at the bottom of each strip.

recorded on the complex prepared as described above. *Figure 3B* (right and left) shows $F_1$-$F_3$ strips from a 3D methyl NOESY spectrum that records correlations of the form ($^{13}C_j$-NOE-$^{13}C_k$-$^1H_k$) with the $F_2$ ($^{13}C_k$) chemical shift from which the slice was taken indicated at the bottom. Each group of three NOESY strips highlights crosspeaks 1–4 from hTRF1 (*Figure 2C*) with correlations 'a'-'d', derived from Ile 401 and Ile 438 of DnaK (*Figure 2E*). Since NOESY crosspeaks connect nearby protons, the residues giving rise to each of the four sets of NOESY triplets (comprising one hTRF1 peak and one each from Ile 401 and Ile 438 of DnaK) all belong to the same conformational state of the hTRF1-DnaK complex, which we refer to as states 'a'-'d' in what follows. In this way hTRF1 conformation 3 (as reported by hTRF1 peak 3 in the $^1H$-$^{13}C$ HMQC of the complex, *Figure 2C*) and DnaK conformation 'd', 2 and 'b', 1 and 'a', and 4 and 'c' are linked. In total four separate hTRF1 residues show sets of crosspeaks to Ile 401 and Ile 438 that report on conformational states 'a'-'d' of DnaK (*Figure 3B*). Thus, at least four methyl-containing residues of hTRF1 can occupy the central position of the DnaK binding cleft (*Figure 3*).

## Assigning the residues of hTRF1 occupying the central position of the DnaK binding cleft

In order to assign the four different peaks from residues at the 0 position (*Figure 3*) we used a mutagenesis-based approach. Initially Leu 30 and Leu 31, at the strongest predicted DnaK binding site on hTRF1, were mutated one at a time to Ile and $^1H$-$^{13}C$ HMQC spectra of the resulting complexes recorded (either ILVM-$^{13}CH_3$ labeled L30I or L31I hTRF1 and U-$^2H$ DnaK/ADP). When Leu 31 was mutated to Ile the peak corresponding to hTRF1 conformation 'a' disappeared from the spectrum (*Figure 4*), thus establishing that peak 'a' arises from a sub-ensemble of DnaK-bound hTRF1 molecules where Leu 31 is at the 0 position. In a similar manner peak 'b' arises from L30 as $L_o$. We subsequently confirmed our assignments via a second approach. A peptide corresponding to hTRF1 residues 23–38 was synthesised with selectively U-$^{13}C$ labeled Leu 30, and the $^1H$-$^{13}C$ HMQC spectrum of this peptide in complex with U-$^2H$ DnaK further established that state 'b' corresponds to residue Leu 30 at the DnaK substrate binding site (*Figure 4—figure supplement 1A*). Moreover, peak 5 of *Figure 2C* could be readily assigned to the second Leu 30 methyl group (*proS*). Similarly, a second peptide, U-$^{13}C$ L31 hTRF1$_{23-38}$ where Leu 31 was U-$^{13}C$, was prepared and used to unambiguously assign state 'a' to the hTRF1-DnaK complex where Leu 31 is located in the binding pocket (*Figure 4—figure supplement 1B*). Peak 6 of *Figure 2C* could be assigned to the *proS* methyl of this residue.

$^1H$-$^{13}C$ HMQC spectra of V-$^{13}CH_3$ hTRF1 in complex with U-$^2H$ DnaK/ADP establish that states 'c' and 'd' correspond to hTRF1 Val residues in the DnaK binding pocket (*Figure 4—figure supplement 2*). Since there are only two Val in hTRF1 (Val 18 and Val 41), each was mutated independently to Ile. The resulting spectra of V18I and V41I hTRF1 in complex with DnaK/ADP (*Figure 4*) lead directly to the assignment of states 'c' and 'd' as arising from Val 41 and Val 18 at position 0, respectively. In addition, the HMQC spectrum of IM-$^{13}CH_3$ DnaK/ADP in complex with the peptide hTRF1$_{34-49}$ (*Figure 2—figure supplement 2*, right) shows distinct peaks at chemical shifts of Ile 401 c and Ile 438 c, confirming the assignment of state c to V41 at position 0. To the best of our knowledge this is the first study where a Val residue has been shown to occupy the 0 position in a DnaK-substrate complex, although Ile, Leu, Met, Pro and cyclohexylalanine are known to be able to bind in the central pocket (*Clerico et al., 2015*).

Having assigned the 'a-d' states of hTRF1 to Leu 31, Leu 30, Val 41 and Val 18, respectively, we next sought assignments for the distinct resonances of Ile 29 (*Figure 2B*). Assignments could not be obtained, however, because NOEs are not observed between Ile 29 and either Leu 30 or Leu 31 in the complex that would immediately link at least some of the multiple Ile correlations with distinct conformations 'a-d'. Although it would be expected that 4 Ile peaks might be observed in the bound state, one for each unique conformation 'a-d', the presence of 5 Ile correlations in the spectrum of *Figure 2B* is of interest. It is unlikely that the additional peak derives from a state whereby Ile 29 is localized to the 0 position of the complex, because NOEs from the γ2 methyl group of Ile 29 and the δ1 methyls of Ile 401 or Ile 438 of DnaK (*Figure 3B*) are not observed, despite the fact that short distances (<5.1 Å to Ile 401 and <4.4 Å to Ile 438) are expected based on an X-ray structure of a peptide complex where Ile occupies the central position (*Zahn et al., 2013*). Some insight into the multiplicity is obtained, however, by inspection of an HMQC spectrum of a complex of IM-$^{13}CH_3$ DnaK/ADP and hTRF1 (*Figure 4—figure supplement 3*) at a low contour level. Notably, two

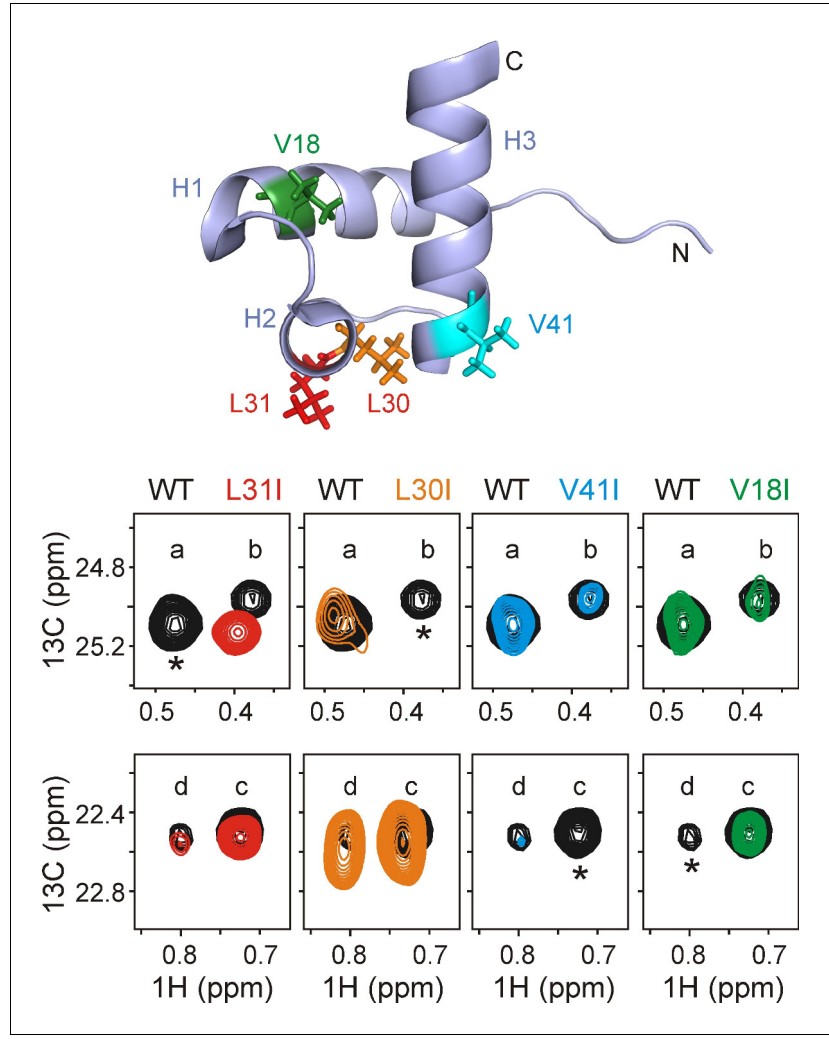

**Figure 4.** Assigning residues of hTRF1 at the central position of the DnaK binding cleft. Selected regions of $^1$H-$^{13}$C HMQC spectra of wild-type ILVM-$^{13}$CH$_3$ hTRF1 bound to U-$^2$H DnaK/ADP (black), overlaid with ILVM-$^{13}$CH$_3$ labeled L31I (red), L30I (orange), V41I (cyan) and V18I (green) hTRF1 bound to U-$^2$H DnaK/ADP. The locations of the four mutations are indicated in the appropriate colour on the structure of native hTRF1 (PDB ID: 1BA5, [**Nishikawa et al., 1998**]) at the top of the figure. The four hTRF1 resonances arising from methyl containing residues at the 0 position of the DnaK binding cavity identified from the 3D NOESY (**Figure 3**) are indicated by letters 'a'-'d'. These four peaks, all assigned to the *proR* $^{13}$CH$_3$ group from *proR* labeling experiments (see text), are highlighted for each mutant considered. One among these four resonances in each column, denoted by an asterisk, disappears in the mutant spectrum thus leading to the assignment of peaks a-d to specific methyl groups in hTRF1.

The following figure supplements are available for figure 4:

**Figure supplement 1.** Assigning states 'a' and 'b' to L31 and L30 respectively.

**Figure supplement 2.** Assigning states 'c' and 'd' to Val residues of hTRF1 present at the 0 position of the DnaK binding cleft.

**Figure supplement 3.** Additional unassigned DnaK-hTRF1 conformations.

additional pairs of unassigned Ile 401 and Ile 438 resonances are observed, indicating the presence of two more states (i.e., beyond the four already characterized). These additional peaks are also present in a complex of hTRF1$_{34-49}$ and DnaK/ADP so that the bound conformers must involve residues within the 34–49 region (see Materials and methods). NOEs correlating the unassigned DnaK Ile 401 and 438 crosspeaks to either of the Val (1) or Leu (1) residues in the 34–49 sequence are not observed; the additional conformations most likely arise, therefore, from other residues at the DnaK 0 position. The appearance of a fifth Ile correlation thus most likely reflects one (or both) of these additional conformers, beyond the four that are characterized presently.

## Quantifying the interaction stoichiometry between hTRF1 and DnaK

In order to estimate the stoichiometries of the diverse complexes that have been characterized above we performed a titration of 0.6 mM Iγ-*proR*-LV-$^{13}$CH$_3$ labeled hTRF1 with increasing amounts of IM-$^{13}$CH$_3$ DnaK/ADP. Intensities of free and bound resonances of both hTRF1 and DnaK in $^1$H-$^{13}$C HMQC spectra were quantified as a function of the increasing [DnaK]$_T$/[hTRF1]$_T$ concentration ratio where the subscript 'T' denotes total concentration (i.e., in both free and bound states). *Figure 5* shows the concentration-dependent intensities of hTRF1 resonances identified to be at the 0

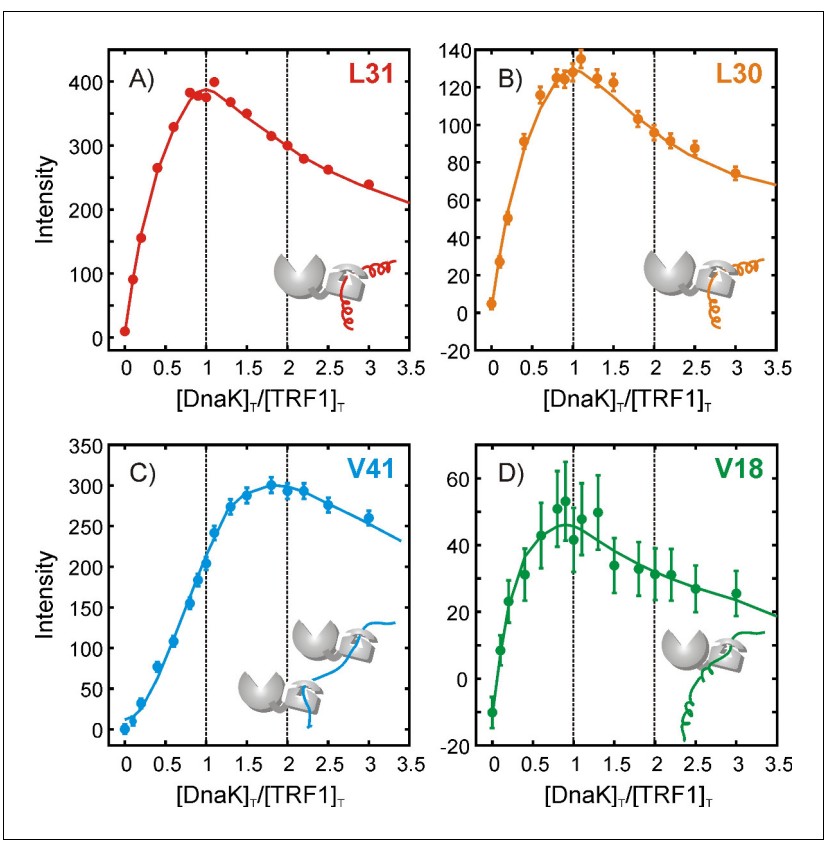

**Figure 5.** Quantifying the stoichiometry of hTRF1 binding to DnaK. Peak intensities from an NMR titration focusing on the four hTRF1 residues Leu 30, Leu 31, Val 18 and Val 41 that occupy the central position of the DnaK binding cavity in different DnaK-bound conformations; (**A**) L31, (**B**) L30, (**C**) V41 and (**D**) V18. The ratio of the total concentrations of DnaK and hTRF1 at each titration point is plotted along the x-axis. Solid lines have been drawn to guide the eye and do not have physical significance.

The following figure supplements are available for figure 5:

**Figure supplement 1.** Simulating multiple binding equilibria for quantifying stoichiometry.

**Figure supplement 2.** Quantifying the stoichiometry of hTRF1 binding to DnaK.

position of the DnaK binding site, corresponding to the four conformations discussed above. As described in Materials and methods the absolute concentration of hTRF1 decreases throughout the course of the titration because the sample for each titration point was generated by removing a calculated volume of solution from the previous titration sample and adding the same volume of a stock solution of DnaK containing no hTRF1.

The mathematical modeling of the titration profiles is complicated by the simultaneous presence of multiple DnaK-hTRF1 binding equilibria in addition to the known oligomerization equilibrium of free DnaK/ADP (*Schönfeld et al., 1995*; *Thompson et al., 2012*) and some aggregation of the sample during the course of the titration. However, general features of the binding equilibria can still be obtained. Numerical simulations applicable to the titration mode employed here show that under conditions of tight binding, species with 1:1 stoichiometry are maximally populated at a $[\text{DnaK}]_T/[\text{TRF1}]_T$ ratio of 1, while species with two bound DnaK molecules occur maximally at a ratio of 2 (Materials and methods, *Figure 5—figure supplement 1*), even when multiple equilibria are present. The maxima for states 'a', 'b' and 'd', corresponding to complexes with Leu 31, Leu 30 and Val 18 occupying position 0, all occur at $[\text{DnaK}]_T/[\text{TRF1}]_T$ ratios of 1, indicating that the stoichiometries in these cases are 1 DnaK to 1 hTRF1 (*Figure 5A,B,D*). Interestingly, however, the maximum for the 'c' state, with Val 41 at 0, is at a ratio of 2, indicating that this sub-ensemble has two DnaK molecules bound to each hTRF1 chain (*Figure 5C*). Currently we have not been able to identify where the second DnaK binds on hTRF1 of conformation 'c'. Similar conclusions are obtained by focusing on cross-peaks from Ile 401 and Ile 438 from the DnaK side as well (*Figure 5—figure supplement 2*). Multiple DnaK molecules have been proposed or modeled to bind large substrate polypeptides such as luciferase (*Szabo et al., 1994*) and rhodanese (*Han and Christen, 2003*; *Kellner et al., 2014*). Our data provides evidence for the existence of higher binding stoichiometries than 1:1 at equilibrium in solution even for a small 53 amino acid substrate, suggesting that multiple occupancy of closely spaced binding sites on typical substrate molecules (*Rüdiger et al., 1997*) is indeed possible. The presence of 2:1 DnaK:hTRF1 complexes amongst at least three other conformers that are singly bound further adds to the heterogeneity of the mixture and the variety of conformations sampled by the hTRF1 chain in the chaperone-bound state.

A qualitative estimate of the relative dissociation equilibrium constants ($K_d$) for the three 1:1 hTRF1-DnaK complexes with Leu 31, Leu 30 and Val 18 occupying position 0 can also be obtained from the titrations, provided differences in relaxation rates for the methyl resonances from these states can be neglected and peak intensities reflect the concentrations of the respective molecular species. In this regard $^1$H and $^{13}$C linewidths of Ile 401 methyl resonances, as measured in $^1$H-$^{13}$C HMQC spectra of hTRF1-bound DnaK in the four states a-d, are similar, with a maximum deviation from the mean of 6.1 Hz in $^1$H and 3.4 Hz in $^{13}$C. This difference in $^1$H linewidth translates roughly into a relative relaxation loss of 13% during constant duration transfer periods in the HMQC pulse sequence, so that differential relaxation losses between the four states are small. With these approximations the intensities of states 'a', 'b' and 'd' at the maxima in the titration profiles provide an estimate of the relative binding strengths. The ratio of Ile 401 resonance intensities for a:b:d at a $[\text{DnaK}]_T/[\text{hTRF1}]_T$ value of 1 are 1:0.4:0.05, corresponding to relative $K_d$ values and free energy differences between the states of 1:2.6:20.1 and 0:0.6:1.8 kcal/mol, respectively, at 35°C. Thus, while each site on hTRF1 binds DnaK with its own specific affinity, and while quantitation is necessarily only approximate, it is clear that the multiple conformations are all thermally accessible and significantly populated at equilibrium.

## Probing rotamer distributions of the DnaK binding site residues

The diversity of substrates that can bind to DnaK implies that the binding pocket possesses considerable flexibility with which to accommodate chains with different sequences and conformations. The binding site residues are also highly conserved in evolution and amino acid substitutions result in impaired function (*Mayer and Bukau, 2005*; *Mayer et al., 2000*), showing that the binding pocket architecture is vital for chaperone activity. In order to estimate the flexibility inherent in the DnaK binding pocket, we analyzed the rotameric distributions of Ile 401 and Ile 438 in the four bound conformations described above. Ile sidechains predominantly sample the trans and gauche- $\chi_2$ rotameric states in solution (*Hansen et al., 2010*) with dihedral angle $\chi_2$ values of 170° and 300° respectively. $^{13}$C chemical shifts are robust reporters of the residue-specific fractions of trans and gauche- (*Hansen et al., 2010*) and using the $^{13}$C δ1 chemical shift values for Ile 401 and Ile 438 in states 'a'-

'd' we computed the fractional populations of the trans and gauche- rotameric states in each case (*Table 1*). The rotamer distributions for the two Ile residues vary substantially in the different bound conformations, ranging from 42% gauche- for Ile 401 and 100% gauche- for Ile 438 in state 'd', to 99% gauche- for Ile 401 and 46% gauche- for Ile 438 in state 'b'. Thus, despite being highly conserved in evolution the DnaK binding site residues retain considerable flexibility, allowing the binding pocket to accommodate different substrate sidechains. Notably, the fractional population of the trans conformer of Ile 438 increases by almost 2-fold, from 28% to 54% between states 'a' and 'b', both of which have a Leu residue at position 0, indicating that it is not merely the identity of the amino acid residing at 0 that governs the topology of the binding cleft.

## Relevance of the observations for other substrates and Hsp70 chaperones

The $\beta$-sandwich domain in the DnaK SBD shows a high degree of conservation within the Hsp70 family of chaperones (*Mayer and Bukau, 2005*; *Zhang et al., 2014*). The conserved topology of the substrate binding site suggests that the multiplicity of DnaK-substrate conformations observed presently should be a common characteristic of Hsp70-client complexes in general. In order to verify this we recorded HMQC spectra of a sample of ILVM-$^{13}CH_3$ hTRF1 containing a two-fold excess of human U-$^2$H Hsc70/ADP (*Figure 6*). The Ile region of this spectrum clearly shows at least four peaks for Ile 29 in the presence of Hsc70, demonstrating the existence of conformational heterogeneity in the Hsc70-hTRF1 complex as well. Three-helix bundle proteins have previously been shown to represent ~10% of all in vivo substrates interacting with *E.coli* DnaK (*Calloni et al., 2012*); in addition, conformational heterogeneity has also been detected in the binding of a $\beta$-sheet protein with DnaK (*Lee et al., 2015*). Taken together, these observations suggest that the promiscuity of Hsp70 and the heterogeneous nature of Hsp70-client complexes are general features, beyond what has been characterized in detail in the present study of the DnaK-hTRF1 complex.

## Discussion

In this study we have used selective isotope-labeling strategies involving methyl group probes in combination with NMR experiments that exploit a methyl-TROSY effect to characterize complexes of the *E.coli* Hsp70 chaperone (DnaK) and a model substrate hTRF1. The beneficial relaxation properties of methyl groups enable the detection of resonances from the DnaK binding site, unlike in $^1$H-$^{15}$N experiments where such resonances are too broad to be observed. We find that the residues that are part of the DnaK binding pocket are inherently malleable so that DnaK can recognize and bind to at least four different sites on the small 6 kDa, 53 residue hTRF1 substrate that have been characterized in the present work. Since typical client proteins of DnaK are much larger than 53 residues, it is very likely that DnaK will bind to multiple client sites, generally. This will lead to a pool of substrates that are segregated into a highly heterogeneous mixture of sub-ensembles, each of which has a DnaK chaperone bound to one or more specific locations, and each with unique conformational features that ultimately facilitate a broad exploration of the free energy landscape (see below).

What advantage does the promiscuity of Hsp70 in client recognition confer on the client protein? Consider a hypothetical substrate with three binding sites for Hsp70 from which three sub-ensembles of Hsp70-bound substrates are generated (*Figure 7*). Chaperone binding results in different lengths of polypeptide both upstream and downstream of the point of contact and the resulting N-

**Table 1.** Percentages of Trans and Gauche- Ile 401 and Ile 438$\chi_2$ Conformations*.

| | a (Leu 31) | | b (Leu 30) | | c (Val 41) | | d (Val 18) | |
| --- | --- | --- | --- | --- | --- | --- | --- | --- |
| | Ile 401 | Ile 438 | Ile 401 | Ile 438 | Ile 401 | Ile 438 | Ile 401 | Ile 438 |
| Trans | 0.86 | 0.28 | 0.99 | 0.54 | 0.66 | 0.00 | 0.58 | 0.00 |
| Gauche- | 0.14 | 0.72 | 0.01 | 0.46 | 0.34 | 1.00 | 0.42 | 1.00 |

*Fractional Trans and Gauche- $\chi_2$ conformations calculated from Ile $^{13}C^{\delta 1}$ chemical shifts according to *Hansen et al. (2010)* for conformers 'a'–'d'. The identities of hTRF1 residues at the 0 position are noted.

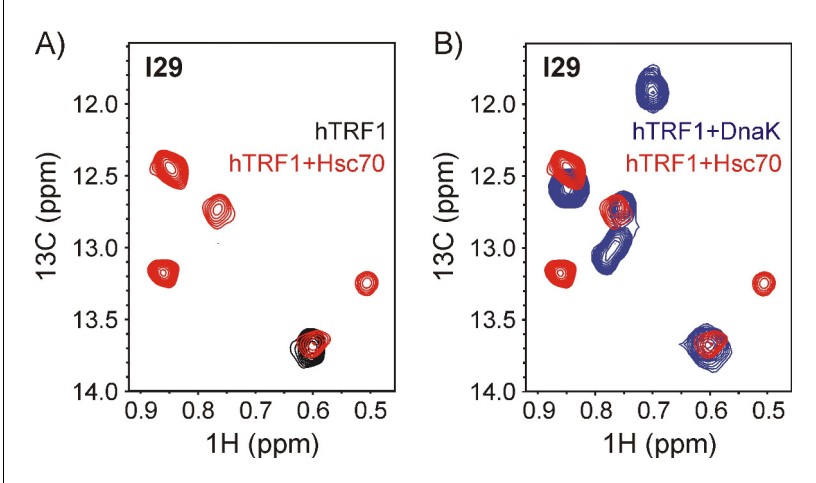

**Figure 6.** Both human Hsc70 and *E.coli* DnaK interact with hTRF1 at multiple sites. Overlay of a $^1$H-$^{13}$C HMQC spectrum of 200 µM ILVM-$^{13}$CH$_3$ hTRF1 containing a two-fold excess of U-$^2$H Hsc70/ADP (red) with the spectrum of (**A**) free ILVM-$^{13}$CH$_3$ hTRF1 (black) and (**B**) ILVM-$^{13}$CH$_3$ hTRF1 bound to U-$^2$H DnaK/ADP (blue). Four peaks are observed for the lone Ile 29 of hTRF1 bound to Hsc70, demonstrating that multiple conformations of the chaperone-substrate complex are formed with both *E.coli* Hsp70 (DnaK) and its human ortholog (Hsc70). DOI: 10.7554/eLife.28030.015

and C-terminal segments, that are different in the three sub-ensembles, can form different amounts of secondary and tertiary structure while bound to Hsp70. Subsequently, when the substrates are released for downstream folding the starting conformations can be significantly different. Some of the starting structures will misfold while others that are more amenable to folding will attain their native form. Misfolded proteins can re-enter the chaperone cycle by binding to Hsp70 using another site, thereby increasing their overall chances of avoiding the same kinetic trap. The heterogeneity resulting from Hsp70 binding thus enables the client protein to explore a number of starting conformations, that lead to the formation of different amounts and types of local structure, as a means for navigating kinetic traps in the conformational free energy landscape and for promoting efficient folding to the native state. Such a mechanism of Hsp70 function is expected to be particularly beneficial for proteins with rugged free energy landscapes and local minima that trap misfolded and aggregation-prone conformations, where folding yield is increased by recycling kinetically trapped molecules at the expense of a decrease in folding rate resulting from repeated binding to the Hsp70 chaperone (*Agashe et al., 2004*). Slower folding in the presence of the Hsp70 chaperone system has indeed been observed in the literature for RNase H (*Sekhar et al., 2012a*), though such a decrease is predicted to be negligible for fast-folding proteins like hTRF1 (56) (*Figure 7—figure supplement 1*).

Our data provides a rationale for why substrate proteins typically undergo multiple rounds of binding and release before folding to the native state (*Mayer, 2013*; *Szabo et al., 1994*; *Clerico et al., 2015*; *Sharma et al., 2010*). Large proteins like luciferase are expected to have several kinetic traps in their folding free energy landscape; in the first binding cycle it is conceivable that only a few sub-ensembles of Hsp70-bound luciferase lead to starting conformations for the client upon release that bypass these kinetic traps and result in efficient folding. The other molecules fall into kinetically trapped misfolded states that then rebind Hsp70. Since only a fraction of folding-competent conformations are generated during any particular iteration of the Hsp70 chaperone cycle, it follows that the net probability of folding and hence the folding efficiency increases with the number of iterations that a substrate undergoes through the chaperone cycle. Our work also sheds light into the cause of the conformational heterogeneity that was previously reported for a complex between the N-terminal SH3 domain from Drk and DnaK based on the observation of multiple peaks for specific residues in a $^1$H-$^{15}$N HSQC spectrum of the complex (*Lee et al., 2015*). While the origin of the peaks was unclear in that study it now appears that DnaK may be able to bind to the SH3 domain in different ways. Multiple modes of binding, including the existence of forward and reverse

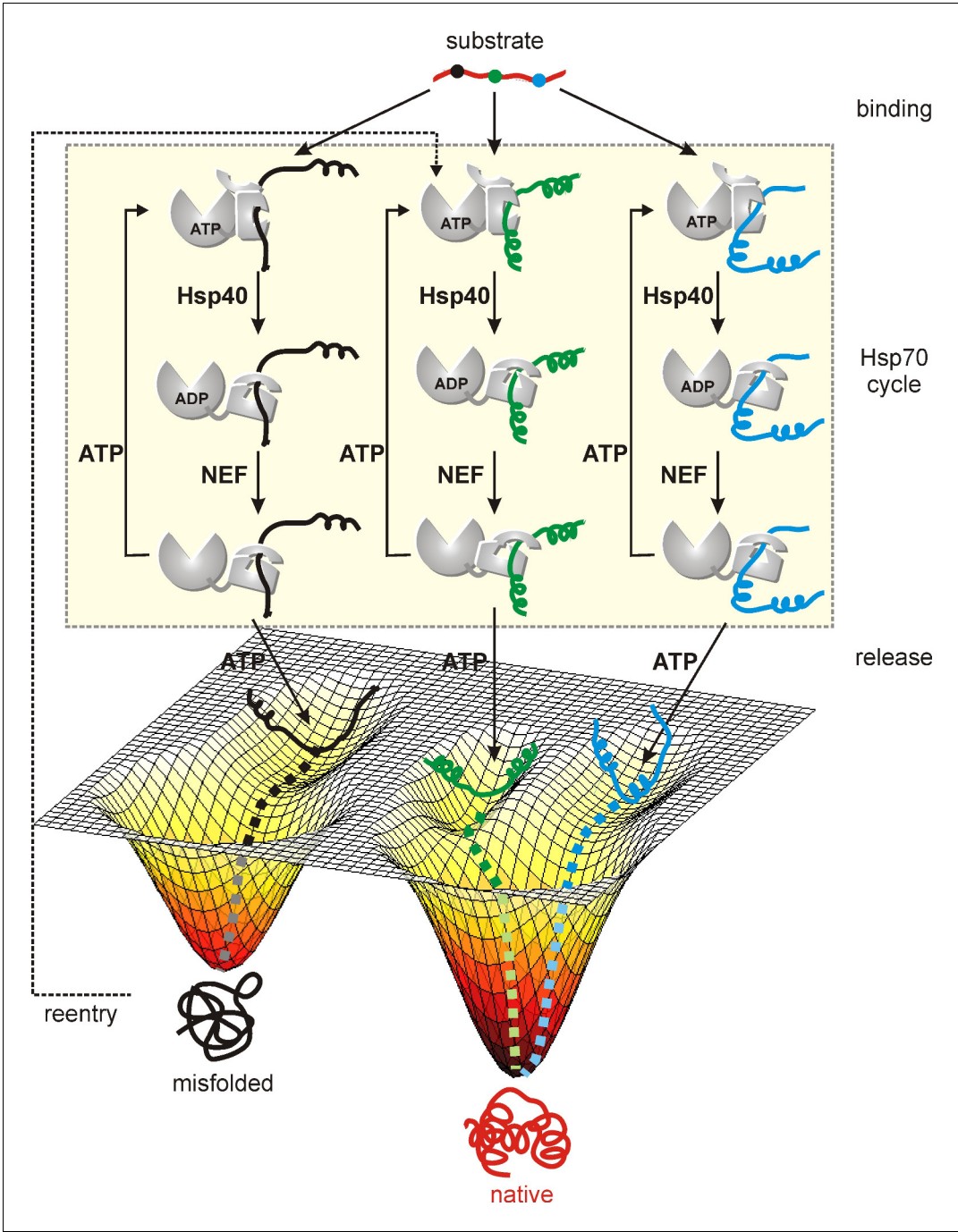

**Figure 7.** Model of a substrate with three Hsp70 binding sites (indicated in different colors) forming three distinct Hsp70 bound sub-ensembles. By increasing the number of interaction sites a larger region of conformational space can be explored from which folding of the substrate occurs upon release from the chaperone. Proteins stuck in kinetic traps can enter the cycle again with Hsp70 binding to different sites increasing the probability of proper folding.

The following figure supplement is available for figure 7:

**Figure supplement 1.** Folding of hTRF1 without and with DnaK/DnaJ/GrpE (K/J/E).

modes and registry shifts, have also been seen in crystal structures of short peptides with DnaK (*Zahn et al., 2013*). Here we establish that such multiple binding modes can exist at equilibrium in solution and we elucidate the molecular determinants for the formation of a heterogeneous chaperone-substrate ensemble.

The model proposed above for Hsp70-mediated protein folding is broadly similar to that proposed for GroEL/ES function, wherein GroEL/ES maintains kinetically trapped conformers in a globally unfolded state and subsequently releases them for folding (*Weissman et al., 1994*; *Todd et al., 1996*). However, in the case of Hsp70 the starting conformation for substrate folding can potentially be different in every binding cycle. Therefore, Hsp70-mediated folding does not rely passively on stochastic refolding from the same conformation, but instead actively alters starting structures as a strategy to sample a different folding pathway and hence different regions of conformational space during each iteration. The presence of multiple chaperone interaction sites on the substrate is a feature shared between Hsp70, SecB (*Lilly et al., 2009*; *Huang et al., 2016*) and trigger factor (*Saio et al., 2014*). However, SecB and trigger factor utilize several surface-exposed hydrophobic pockets for capturing substrate molecules, resulting in a complex where the substrate wraps around the chaperone and buries 4000–8000 $Å^2$ of surface area at the interface (*Lilly et al., 2009*; *Huang et al., 2016*; *Saio et al., 2014*). Such a multivalent mode of interaction is consistent with the role of trigger factor and SecB in keeping the substrate unfolded and extended prior to transfer to downstream Hsp70 chaperones (trigger factor) (*Teter et al., 1999*; *Deuerling et al., 1999*) or in the case of SecB prior to secretion or delivery to the SecA ATPase (*Bechtluft et al., 2010*). In contrast, Hsp70 interacts with substrates using predominantly one binding site (though interactions with the α-helical lid have been observed previously [*Smock et al., 2011*]) that contacts the substrate minimally at 4–5 residues, allowing the rest of the chain to explore conformational space. The mechanism of Hsp70 action is thus different from those of SecB and trigger factor and appears to be tailored towards its function in assisting protein folding.

The chaperone-substrate interaction studied here adds to the increasing number of examples of conformational heterogeneity observed in concert with molecular recognition (*Boehr et al., 2009*; *Mittag et al., 2010*). In the present case the multiplicity of Hsp70 client binding modes leads to static disorder at the binding interface, where each sub-ensemble has a well-defined structure, with dissociation and subsequent re-association of the substrate occurring on a slow timescale. We have shown earlier that parts of the substrate polypeptide chain distal to the Hsp70 binding site are globally unfolded with local secondary structure (*Sekhar et al., 2015*, *2016*), resulting in rapidly interconverting states and dynamic disorder in these regions. Such a fuzzy complex with static and dynamic disorder enhances exploration of the free energy landscape, facilitating the efficient folding of the substrate to its native conformation.

## Materials and methods

### Peptides

Unlabeled hTRF1$_{23-38}$ (EGNWSKILLHYKFNNR) and hTRF1$_{34-49}$ (KFNNRTSVMLKDRWRT) peptides were synthesized by GenScript (http://www.genscript.com). hTRF1$_{23-38}$ labeled with $^{13}C$ at position 30 or 31 was synthesized by New England Peptide (www.newenglandpeptide.com) using U-$^{13}C$-Leu, which was a generous gift from Cambridge Isotopes Limited.

### Protein overexpression, purification and isotope labeling for NMR spectroscopy

#### Hsc70

Human T199A Hsc70$_{1-646}$ (UniProt ID P11142, referred to as Hsc70 in this manuscript) was expressed in *E. coli* BL21-CodonPlus (DE3)-RIPL cells from a pProEx plasmid carrying an N-terminal polyhistidine tag (a gift from J. Young, McGill University, Montreal, Canada). Cells were grown in minimal M9 $D_2O$ media containing 1 g/L $^{15}NH_4Cl$ and 3 g/L [$^2H$,$^{12}C$]-glucose as the sole nitrogen and carbon sources respectively to an $OD_{600}$ of ~0.8, at which point expression was induced by the addition of 0.5 mM isopropyl β-D-1-thiogalactopyranoside (IPTG) and allowed to proceed overnight at 25°C. Following expression, bacteria were harvested and the Hsc70 protein purified on nickel nitrilotriacetic acid (Ni-NTA) resin (GE Healthcare) under denaturing conditions (6 M GuHCl) to ensure complete

removal of bound nucleotides. Hsc70 was refolded on the Ni-NTA column by gradually reducing the concentration of denaturing agent in the wash buffer (from 6 M to 0 M GuHCl), and the N-terminal polyhistidine tag was removed via TEV (tobacco etch virus) protease cleavage. The cleaved protein was further purified on a HiLoad 16/60 Superdex 200 pg gel filtration column (GE Healthcare) equilibrated with 50 mM HEPES (pH 7.5), 50 mM KCl, 1 mM dithiothreitol (DTT), and 0.03% $NaN_3$.

### Mutant and wild-type hTRF1 and DnaK

Samples of hTRF1, both wild-type (wt), and V18I, L30I, L31I and V41I mutants and DnaK, wt and T199A, were prepared and purified as described earlier (*Sekhar et al., 2015*). Proteins were overexpressed in BL21(DE3) cells grown in minimal M9 $D_2O$ media containing 1 g/L $^{15}NH_4Cl$ and 3 g/L [$^2H$,$^{12}C$]-glucose as sole nitrogen and carbon sources respectively. U-$^2H$, Ileδ1-$^{13}CH_3$, Leu/Val-$^{13}CH_3$/$^{12}CD_3$, Met-$^{13}CH_3$-labeling was achieved following the procedures of *Tugarinov et al. (2006)* and *Gelis et al. (2007)*. Stereospecific methyl labeling of Leu and Val prochiral methyls, *proR*-LV-$^{13}CH_3$, was achieved as detailed previously (*Kerfah et al., 2015*; *Gans et al., 2010*), with $^{13}CH_3$-labeling at the Ileγ2 positions using the approach of *Ruschak et al. (2010)*. V-$^{13}CH_3$ hTRF1 was prepared through the addition of the precursor for Leu/Val methyl labeling during cell growth, but with the addition of an excess (100 mg/L) of U-$^2H$,$^{12}C$ Leu (a generous gift from Cambridge Isotopes Limited). NMR samples were prepared in 100% (vol/vol) $D_2O$ buffer (pH 8) composed of 50 mM HEPES, 50 mM KCl, 5 mM $MgCl_2$, 1 mM TCEP, 1 mM EDTA, 0.03% $NaN_3$, and 5 mM ATP or ADP when required.

## NMR spectroscopy

NMR spectra were acquired on 14.1 T Varian INOVA (600 MHz $^1H$ Larmor frequency) or 18.8 T Bruker (800 MHz) spectrometers, equipped with cryogenically cooled probes. All NMR measurements were carried out at 35°C unless specified otherwise. The temperature inside the sample chamber of the spectrometer was measured using a thermocouple inserted into an NMR tube containing $D_2O$. The software packages NMRPipe (*Delaglio et al., 1995*) and Sparky (*Goddard and Kneller, 2006*) were used for processing and visualizing NMR datasets respectively.

## Diffusion measurements

Translational diffusion coefficients were measured on a sample of ILVM-$^{13}CH_3$ labeled hTRF1 (0.3 mM) containing 0.5 mM DnaK/ADP using a 2D $^{13}C$-edited HMQC-based pulsed field gradient (PFG) NMR experiment that is very similar to a previously published $^{15}N$-based sequence but with $^{15}N$ and $^{13}C$ pulses interchanged (*Choy et al., 2002*). Magnetic field gradients were varied between 4 and 60 G/cm with 1 ms encoding and decoding gradient durations, and a diffusion delay of 200 ms was employed. Peak lineshapes were fit using the program FuDA (http://www.biochem.ucl.ac.uk/hansen/fuda/) to extract intensities (I) as a function of squared gradient strength ($G^2$), which were subsequently fit to a single exponential decay function of the form $I = I_0 \exp(-dG^2)$ to determine relative diffusion coefficient values, where $I_0$ is the peak intensity in the absence of encoding/decoding gradients and d is proportional to the diffusion constant.

## $^{13}C$-$^{13}C$-$^1H$ NOESY

Intermolecular NOEs were quantified by recording a 3D $^{13}C$-edited NOESY dataset (200 ms mixing time) from which correlations of the form ($^{13}C_j$-NOE-$^{13}C_k$-$^1H_k$) were obtained. The HMQC version used is based on an HSQC scheme published previously (*Zwahlen et al., 1998*). The NOESY spectrum was acquired on a sample containing 0.7 mM Iγ-*proR*-LV-$^{13}CH_3$ hTRF1 and 1.8 mM IM-$^{13}CH_3$ DnaK/ADP.

## NMR titrations

NMR titrations were carried out starting with a 0.6 mM sample of Iγ-*proR*-LV-$^{13}CH_3$ hTRF1 and a 2.1 mM IM-$^{13}CH_3$ DnaK/ADP stock solution. Titrations were performed by removing an aliquot from the titrating solution and adding an identical volume of the stock DnaK/ADP solution. Consequently, the concentration of hTRF1 decreases during the course of the titration while the volume of the sample remains constant. Peak lineshapes were fit globally across all titration points using FuDA to estimate intensity as a function of the $[DnaK]_T$ / $[hTRF1]_T$ ratio.

## Simulating NMR titration profiles

Consider the binding of DnaK ($K$) to hTRF1, which in turn exchanges between native ($N$) and unfolded ($U$) states at equilibrium, with a folding constant $K_{UN} = \frac{[N]}{[U]}$ (the notation [$i$] stands for the equilibrium concentration of species $i$). We have previously measured $K_{UN}$ to be 22.7, 35°C, based on $^{15}$N CEST experiments (**Vallurupalli et al., 2012**). In the models that follow we assume that $K$ binds $U$ and not $N$, as we have assumed previously (**Sekhar et al., 2015**), although this is not essential for any of the conclusions presented. Let the total concentration of DnaK and hTRF1 at a titration point be $[K]_T$ and $[N]_T$ respectively, where we note that successive titrations points involve the removal of a given amount of solution and replacing it with an equivalent volume of DnaK, as described in 'NMR titrations' above. Consider, first, the formation of both 1:1 ($UK$) and 1:2 ($UK2$) hTRF1-DnaK complexes with association (dissociation) constants $K_1$ ($K_{d1}$) and $K_2$ ($K_{d2}$) such that $K_1 = \frac{1}{K_{d1}} = \frac{[UK]}{[U][K]}$ and $K_2 = \frac{1}{K_{d2}} = \frac{[UK2]}{[UK][K]} = \frac{[UK2]}{K_1[U][K]^2}$, Linear Model of **Figure 5—figure supplement 1**. The mass balance equations are given by

$$[N]_T = [N] + [U] + [UK] + [UK2],$$

$$[K]_T = [K] + [UK] + 2[UK2].$$

Taken together these equations can be used to derive an expression for the free DnaK concentration at equilibrium as the solution of the cubic equation

$$a[K]^3 + b[K]^2 + c[K] + d = 0,$$

where

$$a = K_1 K_2,$$

$$b = K_1 + K_1 K_2 \big(2[N]_T - [K]_T\big),$$

$$c = 1 + K_{UN} + K_1 \big([N]_T - [K]_T\big),$$

$$d = -(1 + K_{UN})[K]_T.$$

The cubic equation was solved using the software package MATLAB (The Mathworks Inc., Natick, MA, U.S.A.) and combined with the expressions for equilibrium constants to yield concentrations of the various species at each of the $[K]_T$ and $[N]_T$ values used in the titration. These in turn form the basis for the Linear Model of **Figure 5—figure supplement 1**.

We have also considered a more complex model (Square Model of **Figure 5—figure supplement 1**) in which there are two binding sites for DnaK resulting in the formation of 1:1 complexes $UKA$ and $UKB$, with association (dissociation) constants $K_1$ ($K_{d1}$) and $K_2$ ($K_{d2}$) such that $K_1 = \frac{1}{K_{d1}} = \frac{[UKA]}{[U][K]}$ and $K_2 = \frac{1}{K_{d2}} = \frac{[UKB]}{[U][K]}$. $UKA$ and $UKB$ can then bind a second molecule of DnaK to form a dimer $UKAB$. The two binding sites are assumed to be independent of each other, and consequently the opposite arms of the square model have identical equilibrium constants. This gives an expression for the equilibrium concentration of the dimeric species $UKAB$ as

$$[UKAB] = K_2[UKA][K] = K_1[UKB][K] = K_1 K_2[U][K]^2.$$

The mass balance equations are given by

$$[N]_T = [N] + [U] + [UKA] + [UKB] + [UKAB],$$

$$[K]_T = [K] + [UKA] + [UKB] + 2[UKAB].$$

The expression for the free DnaK concentration at equilibrium for the Square Model of **Figure 5—figure supplement 1** is the solution of the cubic equation

$$a[K]^3 + b[K]^2 + c[K] + d = 0,$$

where

$$a = K_1 K_2,$$

$$b = K_1 + K_2 + K_1 K_2 \big(2[N]_T - [K]_T\big),$$

$$c = 1 + K_{UN} + (K_1 + K_2)\big([N]_T - [K]_T\big),$$

$$d = -(1 + K_{UN})[K]_T.$$

The cubic equation was solved using MATLAB and combined with the expressions for equilibrium constants to yield concentrations of the various species at each of the $[K]_T$ and $[N]_T$ values used in the titration.

## Acknowledgements

This work was supported by grants from the Canadian Institutes of Health Research and the Natural Sciences and Engineering Research Council of Canada (LEK) as well as funding from the Azrieli Foundation, the Blythe Brenden-Mann New Scientist Fund, and the Roshan Family Foundation (RR). LEK holds a Canada Research Chair in Biochemistry. The authors are grateful to Professor Lila Gierasch, University of Massachusetts, for generously providing partial methyl assignments for DnaK/ADP.

## Additional information

### Competing interests

LEK: Reviewing editor, *eLife*. The other authors declare that no competing interests exist.

### Funding

| Funder | Grant reference number | Author |
|---|---|---|
| Canadian Institutes of Health Research | | Lewis E Kay |
| Natural Sciences and Engineering Research Council of Canada | | Lewis E Kay |
| Azrieli Foundation | | Rina Rosenzweig |
| Blythe Brenden-Mann Foundation | New Scientist Fund | Rina Rosenzweig |
| Roshan Family Foundation | | Rina Rosenzweig |

The funders had no role in study design, data collection and interpretation, or the decision to submit the work for publication.

### Author contributions

RR, AS, Conceptualization, Data curation, Formal analysis, Investigation, Visualization, Methodology, Writing—original draft, Writing—review and editing; JN, Resources, Formal analysis, Methodology; LEK, Conceptualization, Data curation, Formal analysis, Supervision, Funding acquisition, Investigation, Visualization, Methodology, Writing—original draft, Writing—review and editing

### Author ORCIDs

Ashok Sekhar, http://orcid.org/0000-0002-8628-7799
Lewis E Kay, http://orcid.org/0000-0002-4054-4083

## Additional files

### Major datasets

The following previously published datasets were used:

| Author(s) | Year | Dataset title | Dataset URL | Database, license, and accessibility information |
|---|---|---|---|---|
| Nishikawa T, Nagadoi A, Yoshimura S, Aimoto S, Nishimura Y | 1998 | DNA-BINDING DOMAIN OF HUMAN TELOMERIC PROTEIN, HTRF1, NMR, 18 STRUCTURES | http://www.rcsb.org/pdb/explore/explore.do?structureId=1ba5 | Publicly available at the RCSB Protein Data Bank (accession no: 1BA5) |
| Zuiderweg ERP, Bertelsen EB | 2009 | NMR-RDC / XRAY structure of E. coli HSP70 (DNAK) chaperone (1-605) complexed with ADP and substrate | http://www.rcsb.org/pdb/explore/explore.do?structureId=2kho | Publicly available at the RCSB Protein Data Bank (accession no: 2KHO) |
| Zhu X, Zhao X, Burkholder WF, Gragerov A, Ogata CM, Gottesman ME, Hendrickson WA | 1996 | THE SUBSTRATE BINDING DOMAIN OF DNAK IN COMPLEX WITH A SUBSTRATE PEPTIDE, DETERMINED FROM TYPE 1 NATIVE CRYSTALS | http://www.rcsb.org/pdb/explore/explore.do?structureId=1dkz | Publicly available at the RCSB Protein Data Bank (accession no: 1DKZ) |

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
