## [Decision Letter]

Thank you for submitting your article "Promiscuous binding by Hsp70 results in conformational heterogeneity and fuzzy chaperone-substrate ensembles" for consideration by *eLife*. Your article has been reviewed by three peer reviewers, one of whom, Volker Dötsch (Reviewer #1), is a member of our Board of Reviewing Editors and the evaluation has been overseen by John Kuriyan as the Senior Editor.

The reviewers have discussed the reviews with one another and the Reviewing Editor has drafted this decision to help you prepare a revised submission. All reviewers liked the manuscript and the conclusions and had only a few comments to be addressed in a revised version of the manuscript.

Summary:

Hsp70 is a central hub for chaperones and it helps with protein folding, translocation and disaggregation. The mechanistic basis for Hsp70 chaperone activity remains still poorly understood because of the difficulty in characterizing the dynamic and heterogeneous complex with substrate proteins. Previous Hsp70-client studies mostly used short peptides as substrates, an over-simplified scenario given that physiological clients typically have hundreds of residues. In this paper the authors took advantage of Methyl-TROSY NMR methods to dissect the interaction between *E. coli* Hsp70 DnaK and a model substrate hTRF1. The HMQC spectra indicate that there are multiple DnaK-bound conformations. DnaK binds to four sites on hTRF1 evidenced by intermolecular NOEs. This is a very well designed and solid paper that provides important mechanistic insights into the Hsp70 chaperone-client interaction modes and how Hsp70 can modulate the folding landscape of clients.

Essential revisions:

1) The authors use HMQC spectra of hTRF1 bound to DnaK/ATP to rule out the lid being the source of the observed conformational heterogeneity. The assumption here is that in this case the lid is docked to NBD. However previous work (Cell 151, 1296-1307, December 7, 2012) showed that in the ATP and substrate binding state Hsp70 is in a lid-undocked and linker-bound state. The authors possibly need to revise the text?

2) The authors probed the rotamer distribution of the DnaK binding site residues to explore mechanisms of promiscuity of DnaK binding. In subsection “Probing rotamer distributions of the DnaK binding site residues”, the authors argued that the residues in binding site are highly conserved and thus indicated that the binding pocket is rigid. Is this always true though? Does sequence conservation necessarily mean structural rigidity?

3) Given the availability of structural data on TF and SecB chaperones in complex with non-native proteins, it would be beneficial to the field if a comparison of the different binding modes (Hsp70, TF, SecB) were included in the current work.

4) There are 4 binding sites on hTRF1, however in Figure 2, there are 5 extra peaks for I29, Since this spectra acquired at a ratio of 2 for DnaK/hTRF1, does this mean that V41 at P0 position has two states, stoichiometry 1 and 2? Can the authors clarify in the text?

5) The long interaction time of the chaperone with the unfolded protein and the possibility of multiple binding events can certainly help proteins that fold slowly. However, the model would also predict that the folding of fast and easy folding proteins should be inhibited and slowed down. Has this been observed?

---

## [Author Response]

*1) The authors use HMQC spectra of hTRF1 bound to DnaK/ATP to rule out the lid being the source of the observed conformational heterogeneity. The assumption here is that in this case the lid is docked to NBD. However previous work (Cell 151, 1296-1307, December 7, 2012) showed that in the ATP and substrate binding state Hsp70 is in a lid-undocked and linker-bound state. The authors possibly need to revise the text?*

We have now modified the text to incorporate this suggestion in subsection “Characterizing the multiple conformations of the DnaK-hTRF1 complex” of the revised manuscript.

"Next, we tested whether the multiple conformations arise from the interactions between substrate and the α-helical chaperone lid that encloses the substrate-binding pocket. […] Substrate binding to DnaK/ATP results in the formation of an allosterically active DnaK ensemble that consists of domain-docked as well as -undocked conformations (Zhuravleva, Clerico and Gierasch, 2012) in which the helical lid is more mobile than in the DnaK/ADP state, as inferred from faster substrate release rates in the ATP-bound conformation (Mayer and Bukau, 2005 and Gisler, Pierpaoli and Christen, 1998)."

*2) The authors probed the rotamer distribution of the DnaK binding site residues to explore mechanisms of promiscuity of DnaK binding. In subsection “Probing rotamer distributions of the DnaK binding site residues”, the authors argued that the residues in binding site are highly conserved and thus indicated that the binding pocket is rigid. Is this always true though? Does sequence conservation necessarily mean structural rigidity?*

The reviewer brings up a good point and indeed sequence conservation does not necessarily mean rigidity. We have now modified this sentence to remove the suggestion that conservation in sequence implies structural rigidity as follows (Subsection “Probing rotamer distributions of the DnaK binding site residues” of the revised manuscript):

"The diversity of substrates that can bind to DnaK implies that the binding pocket possesses considerable flexibility with which to accommodate chains with different sequences and conformations. The binding site residues are also highly conserved in evolution and amino acid substitutions result in impaired function (Mayer and Bukau, 2005 and Mayer et al., 2000), showing that the binding pocket architecture is vital for chaperone activity."

*3) Given the availability of structural data on TF and SecB chaperones in complex with non-native proteins, it would be beneficial to the field if a comparison of the different binding modes (Hsp70, TF, SecB) were included in the current work.*

We have now added the following paragraph in the Discussion section of the manuscript that juxtaposes the mode of action of Hsp70 with that of trigger factor and SecB.

"The presence of multiple chaperone interaction sites on the substrate is a feature shared between Hsp70, SecB (Lilly, Crane and Randall, 2009 and Huang et al., 2016) and trigger factor (Saio et al., 2014). […] The mechanism of Hsp70 action is thus different from those of SecB and trigger factor and appears to be tailored towards its function in assisting protein folding."

*4) There are 4 binding sites on hTRF1, however in Figure 2, there are 5 extra peaks for I29, Since this spectra acquired at a ratio of 2 for DnaK/hTRF1, does this mean that V41 at P0 position has two states, stoichiometry 1 and 2? Can the authors clarify in the text?*

This is an interesting question for which we do not have a conclusive answer. The main problem is that the Ile 29 peaks in Figure 2 cannot be assigned to conformations 'a' – 'd' of the DnaK-hTRF1 complex, because there are no NOEs from Ile 29, either to Ile 401/Ile 438 in DnaK, or to Leu 30 or Leu 31 in hTRF1.

In order to address the question of why there are five Ile 29 peaks for four conformations, we have revisited the HMQC spectrum of the peptide containing the V41 binding site, hTRF1_34-49_, in complex with IM-^[12]^CH_3_ DnaK/ADP. In addition to Ile 401c and Ile 438c expected from this peptide two additional unassigned Ile 401/438 pairs are observed (now illustrated in Figure 4—figure supplement 3). These peaks are also present in the HMQC spectrum of full-length hTRF1 bound to IM-^[12]^CH_3_ DnaK (Figure 4—figure supplement 3) and must arise from non-ILV residues binding to the 0 position, because these Ile 401/438 peaks do not show NOEs to any hTRF1 Leuδ1, Valγ1 or Ileγ2 groups. Thus, while we have assigned four conformations of the DnaK-hTRF1 complex, it is clear that there are at least two more, accounting for the greater-than-four Ile 29 peaks in Figure 2.

We also cannot rule out the possibility raised by the reviewers that the multiplicity in the Ile region (Figure 2) could be also the result of distinct chemical shifts for Ile 29 in the monomeric and dimeric forms of state 'c'. Indeed, we cannot also eliminate the possibility that one of the weak peaks in this region arises from the small fraction of unfolded hTRF1 at equilibrium, which would be expected to resonate in the same region of the HMQC.

Taken together it is virtually certain that there are at least six conformations of the DnaK-hTRF1 complex. However, we have not included a detailed discussion of the unassigned states in the manuscript, as we would like to base our conclusions on the peaks we can assign with certainty. We have added a section in Results addressing the question of why there are more than four Ile 29 peaks:

"Having assigned the 'a-d' states of hTRF1 to Leu 31, Leu 30, Val 41 and Val 18, respectively, we next sought assignments for the distinct resonances of Ile 29 (Figure 2).[…] The appearance of a fifth Ile correlation thus most likely reflects one (or both) of these additional conformers, beyond the four that are characterized presently. "

*5) The long interaction time of the chaperone with the unfolded protein and the possibility of multiple binding events can certainly help proteins that fold slowly. However, the model would also predict that the folding of fast and easy folding proteins should be inhibited and slowed down. Has this been observed?*

The interaction time of DnaK with substrate is long only in the DnaK/ADP state. in vivo there is more ATP than ADP and the affinity of DnaK is higher for ATP than ADP, making ATP-DnaK the major nucleotide state of the chaperone (Sekhar, Lam and Cavagnero, 2012). Since ATP-DnaK interacts only transiently with substrates, the interaction times are not as long as in the DnaK/ADP state and a typical binding/release cycle for a substrate takes ~ 1 s (Mayer and Bukau, 2005).

Our model does predict that folding-competent proteins will be slowed down by interaction with the Hsp70 chaperone system and this has indeed been observed for RNase H (Sekhar et al., 2012, folding rate constant k_f_ = 0.03 s^-1^). It is also known that DnaK optimizes the folding yield at the expense of the folding rate, and that the co-translational folding of luciferase is slowed in the presence of DnaK/DnaJ/GrpE, because the nascent luciferase chain binds DnaK (Agashe et al., 2004). The decrease in folding speed is not expected to be significant for a fast-folding protein like hTRF1 where folding is faster than the binding event. We have simulated the folding of hTRF1 without and with DnaK/DnaJ/GrpE/ATP and find that the rate constants of folding in the two cases are virtually identical (new Figure 7—figure supplement 1). We have also modified the text under Discussion as:

"Such a mechanism of Hsp70 function is expected to be particularly beneficial for proteins with rugged free energy landscapes and local minima that trap misfolded and aggregation-prone conformations, […] though such a decrease is predicted to be negligible for fast-folding proteins like hTRF1 (Sekhar et al., 2012) (Figure 7—figure supplement 1)."